# Deep Sea Minerals Ameliorate Dermatophagoides Farinae- or 2,4-Dinitrochlorobenzene-Induced Atopic Dermatitis-like Skin Lesions in NC/Nga Mice

**DOI:** 10.3390/biomedicines13040861

**Published:** 2025-04-02

**Authors:** Hyo Sang Kim, Myeong Hwan Kim, Byeong Yeob Jeon, You Kyung Jang, Jeong Ki Kim, Hyun Keun Song, Kilsoo Kim

**Affiliations:** 1College of Veterinary Medicine, Kyungpook National University, Daegu 41566, Republic of Korea; poppingboys@naver.com (H.S.K.); meong125678@naver.com (M.H.K.); 2Qualified Bio & Minerals Co., Ltd., Seoul 06752, Republic of Korea; byjeon01@qbm.co.kr (B.Y.J.); ykjang01@qbm.co.kr (Y.K.J.); 3MEDI Co., Ltd., Okcheon-eup 29040, Chungcheongbuk-do, Republic of Korea; drks1kim@gmail.com

**Keywords:** deep sea mineral, inflammation, IgE, anti-atopic dermatitis

## Abstract

**Background**: Chronic pruritus and inflammatory skin lesions, characterized by high recurrence, are hallmarks of atopic dermatitis (AD). Despite its increasing prevalence, the development of therapeutic agents for AD remains limited. This study aimed to evaluate the therapeutic effects of deep sea minerals (DSMs) in mist and cream formulations on the development of AD-like skin lesions in NC/Nga mice exposed to either *Dermatophagoides farinae* body extract (Dfb) or 2,4-dinitrochlorobenzene (DNCB). **Methods**: To induce AD, 100 mg of Biostir AD cream containing crude Dfb or 200 µL of DNCB (1%) was topically applied to the dorsal skin of NC/Nga mice. Additionally, 200 µL of deep sea mineral mist (DSMM) and 10 mg of deep sea mineral cream (DSMC) were applied daily to the dorsal skin for 4 weeks. AD was assessed through visual observations, clinical scoring of skin severity, serological tests, and histological analysis. **Results**: Visual and clinical evaluations revealed that DSMs inhibited the formation of AD-like skin lesions. DSMs also significantly affected trans-epidermal water loss and erythema. Treatment with DSMs resulted in reduced serum levels of IgE, IFN-γ, and IL-4. Histological analysis indicated that DSMs decreased skin thickness. Immunostaining for the CD4 antigen demonstrated a reduced infiltration of CD4^+^ T cells, which drive the Th2 response in AD, following DSM treatment. **Conclusions**: In conclusion, the cream formulation of DSMs showed better results than the mist formulation. These results suggest that DSMs may be an effective treatment for AD-like skin lesions, especially in cream formulation.

## 1. Introduction

Atopic dermatitis (AD) is a chronic inflammatory skin condition characterized by symptoms such as edema, vesicles, and weeping during the acute phase, as well as increased skin thickness and hypersensitivity to various antigens in the chronic phase. The underlying causes of AD include genetics, immune dysregulation, epidermal dysfunction, and environmental factors [1,2]. AD affects 10% to 20% of the global population, and its prevalence is rapidly increasing. Recent studies indicate a steady rise in allergic diseases among children, with 47.2% of preschoolers, 26% of elementary school students, and 17.5% of middle school students diagnosed with AD [3,4]. While there is currently no definitive cure for AD, three therapeutic approaches—anti-inflammatory drugs, skin barrier reconstitution creams, and physical treatments—are utilized to alleviate severe symptoms. Anti-inflammatory medications, such as corticosteroids and calmodulin inhibitors, help manage AD by suppressing various immune responses. However, the long-term use of these medications carries significant risks and requires careful monitoring [5,6].

Nishiki-nezumi Cinnamon/Nagoya (NC/Nga) mouse models, known for their spontaneous development of skin lesions resembling those observed in human AD, typically begin to exhibit symptoms around 8 weeks of age under conventional conditions [7,8]. Various methods have been identified to accelerate the onset of disease in NC/Nga mouse models maintained under specific pathogen-free (SPF) conditions, including the application of allergens or chemicals. The *Dermatophagoides farinae* body (Dfb)-induced AD model involves the repetitive application of Dfb extract, which is the most prevalent mite found on the skin of AD patients and serves as one of the strongest allergic antigens associated with AD [9,10,11,12]. Consequently, AD-like mouse models that closely mimic the human condition may be valuable for elucidating the pathogenesis of AD. The chemical induction method employing 2,4-dinitrochlorobenzene (DNCB), an electrophilic and cytotoxic derivative of benzene, also induces AD-like skin disease in NC/Nga mice [13,14]. The AD-like skin disorders in these mice are characterized by behaviors such as scratching, the rapid development of erythema, lichenification due to edema, and hemorrhage [15,16]. Histological analysis reveals eosinophilia, along with mast cell hyperplasia and dense accumulation in the skin lesions. Furthermore, serological analysis indicates elevated total serum levels of immunoglobulin E (IgE) and AD-related cytokines, including interleukin (IL)-4 and interferons [14,16,17,18].

Sea water obtained from depths greater than 200 m is referred to as deep sea water. It is characterized by its high purity, rich nutritional content, stability, and low temperature, making it a valuable raw material in the food and pharmaceutical industries. Research has indicated that deep sea water exhibits natural anti-diabetic, anti-cholesterol, anti-obesity, and anti-cancer properties [19,20,21,22]. These beneficial effects are attributed to the presence of essential minerals such as calcium (Ca), magnesium (Mg), sodium (Na), and zinc (Zn) [19]. Several studies have shown that deep sea water can help alleviate inflammation associated with skin diseases; in particular, magnesium has been reported to inhibit inflammatory cytokines [23,24].

Consequently, these findings suggest that the minerals present in deep sea water may function as natural anti-inflammatory agents for managing the progression of AD. However, the precise regulatory mechanisms involved remain unclear. In this study, mist and cream formulations of deep sea minerals (DSMs) known to have anti-inflammatory properties were evaluated for their efficacy in alleviating or improving allergic dermatitis in two types of AD-like mouse models.

## 2. Materials and Methods

### 2.1. Animals

Thirty-five SPF 6-week-old male NC/Nga mice were obtained from Central Lab Animal Inc. (Seoul, Republic of Korea) and acclimated for one week prior to the experiment. The mice were housed individually in ventilated cages within an animal room that maintained controlled environmental conditions (12-h light/dark cycle, 22 ± 1°C, and 50 ± 10% relative humidity). They were provided with a standard laboratory diet (laboratory chow 5057, Purina Korea, Seongnam, Republic of Korea) and had access to water ad libitum in the SPF facility. All animal protocols were approved by the Committee for Animal Experiments at Kyungpook National University (IACUC, KNU 201762). The deep sea mineral cream (DSMC) and deep sea mineral mist (DSMM) used in this experiment were provided by Qualified Bio & Minerals Co (Seoul, Republic of Korea). The deep sea mineral cream and deep sea mineral mist contain 5% deep sea minerals.

### 2.2. Induction of Atopic Dermatitis and DSM Application

Two substances were used to experimentally induce AD: DNCB and house dust mite antigen. The seven groups were treated according to their respective protocols. After one week of adaptation, the mice were randomly assigned to the following groups: Group 1: untreated normal control (NC); Group 2: Dfb-induced AD as a positive control (Dfb control); Group 3: Dfb + DSMM (Dfb mist); Group 4: Dfb ointment + DSMC (Dfb cream); Group 5: DNCB-induced AD as a positive control (DNCB control); Group 6: DNCB + DSMM (DNCB mist); Group 7: DNCB + DSMC (DNCB cream).

AD was induced by applying Biostir AD (Biostir Inc., Kobe, Japan), a natural chemical derived from dust mites, to the skin of the mice for a duration of 4 weeks. This reagent, composed of dust mite allergens (*Dermatophagoides farinae*), elicits subclinical symptoms that resemble those of atopy in humans. To induce AD, 100 mg of Biostir AD cream containing crude Dfb was applied to the shaved dorsal skin of the mice. On the eighth day following the initial Dfb application, DSMs were administered daily for 4 weeks.

To induce various types of AD-like skin lesions, the animals’ skin was shaved with animal clippers on day 6. On day 7, 200 µL of 1% DNCB, dissolved in a mixture of acetone and olive oil (3:1, *v*/*v*), was applied to the dorsal skin. For a duration of 4 weeks, 200 µL of DSMM and 10 mg of DSMC were applied daily to the dorsal skin. The dosages of DSMM and DSMC were selected based on a safe concentration that was not identified as irritating in an initial test, where albino rabbits were exposed to a skin area of 6 cm^2^ for 4 h.

Mice were euthanized through CO_2_ inhalation following a 4-week treatment with DSMs. During the autopsy, blood was collected from the posterior vena cava, and skin samples were excised from the dorsal region of the mice for further analysis.

### 2.3. Evaluation of Skin Lesion Severity

The lesions on the dorsal skin were macroscopically assessed based on the following symptoms: erythema/hemorrhage, edema, scarring/dryness, and excoriation/erosion. The total clinical skin score for the AD-like lesions on the NC/Nga mice was defined as the sum of individual scores, which were graded as 0 (none), 1 (mild), 2 (moderate), or 3 (severe), with a possible range from 0 to 12 [7]. At the beginning of this experiment, the score for each group was recorded as 0. Subsequently, the severity of dermatitis on the skin lesions was assessed twice a week. These visual evaluations were conducted by at least two independent investigators. Changes in the skin symptoms of the NC/Nga mice were documented using photographs. The trans-epidermal water loss (TEWL) of the dorsal skin was measured 4 weeks after the application of DSMs using VapoMeter (Delfin Technologies Ltd., Kuopio, Finland).

### 2.4. Measurement of Serum IgE, IL-4, and Interferon Gamma (IFN-γ) Levels

Mice were anesthetized through inhalation of isoflurane (2–2.5%), and blood was immediately collected from the posterior vena cava. The blood samples were centrifuged at 2000× *g* for 20 min at 4 °C. Serum was then collected and stored at −70 °C until analysis. Total serum IgE concentrations were quantified using a mouse IgE enzyme-linked immunosorbent assay (ELISA) kit (Shibayagi Ltd., Shibukawa, Gunma Pref., Japan) and a mouse IFN-γ and IL-4 ELISA kit (R&D Systems Inc., Minneapolis, MN, USA), following the manufacturer’s instructions.

### 2.5. Histologic Analysis and Immunohistochemistry

Mice were euthanized after the experiment was completed. The dorsal skin was excised, fixed in 10% neutral buffered formalin, embedded in paraffin, sectioned into 10 µm slices, stained with hematoxylin and eosin (H&E) solution, and subsequently examined using light microscopy. Based on the histological findings, a grade of severe dermatitis was assessed in a blinded manner, revealing the infiltration of inflammatory cells in the corium and hyperkeratosis, hypertrophy, and inflammatory cell infiltration in the epidermis.

For immunohistochemistry, the paraffin-embedded tissue sections on slides were deparaffinized. The sections were incubated with rabbit anti-CD4 (Abcam, Cambridge, UK) overnight at 4 °C, washed with PBS containing 0.05% Tween 20, and then incubated with biotin-conjugated goat anti-rabbit IgG was used as secondary antibody (1:100 in TBS; Dako., Carpinteria, CA, USA), 3,39-diaminobenzidine tetrahydrochloride (DAB, brown) (Sigma-Aldrich, St. Louis, MO, USA). Images of each section were obtained under a microscope (Nikon E600; Tokyo, Japan).

### 2.6. Statistics

All data are expressed as the mean ± standard error of the mean (SEM). Two-tailed Student’s *t*-tests were employed to assess statistical significance for the differences in means, with significance set at *p* < 0.05. The significance and validity of the data were further confirmed by conducting a one-way analysis of variance (ANOVA).

## 3. Results

### 3.1. Treatment with DSMs Alleviates Dfb- or DNCB-Induced AD-like Skin Lesions in NC/Nga Mice

The therapeutic efficacy of DSM mist and cream formulations on the development of AD-like skin lesions was evaluated. Figure 1 shows the macroscopic evaluation after 4 weeks of application of DSM mist and cream formulations. As illustrated in Figure 1A, the clinical characteristics of AD-induced NC/Nga mice were assessed by shaving the hair from the back of the mice and photographing the exposed area. When Dfb or DNCB were used to induce AD, the mice exhibited thickened skin along with symptoms of severe erythema, hemorrhage, edema, scarring, erosion, and excoriation. Treatment with DSMs effectively inhibited the induction of these AD-like skin lesions. Both the mist and cream formulations of DSMs demonstrated inhibitory effects compared to the control group, with the cream formulation showing a more pronounced inhibitory effect. Clinical severity scores for each of the four symptoms of AD were evaluated and are presented in Figure 1B. The skin severity score was significantly lower in the DSM-treated group than in the control AD mice (*p* < 0.005). Additionally, the cream formulation resulted in a lower skin severity score than the mist formulation.

### 3.2. Treatment with DSMs Inhibits Skin Dehydration and Increases in Erythema Thickness in Dfb- or DNCB-Induced AD-like Skin Lesions in NC/Nga Mice

To investigate the effects of DSMs on water loss and erythema thickness in NC/Nga mouse skin, trans-epidermal water loss (TEWL) and erythema thickness were evaluated after 4 weeks of DSM application. TEWL was significantly increased in the Dfb- or DNCB-treated groups compared to the naïve control group. In contrast, TEWL was reduced in the DSM-treated groups (Figure 2A). Furthermore, the TEWL of mice treated with the cream formulation of DSMs was lower than that of those treated with the mist formulation in Dfb- or DNCB-induced AD mice. Similarly, erythema thickness was decreased in the DSM-treated groups compared to the Dfb- or DNCB-treated groups, with the cream formulation group exhibiting lower erythema thickness levels than the mist formulation group (Figure 2B).

### 3.3. Treatment with DSMs Downregulates Dfb- or DNCB-Induced Serum IgE, IL-4, and IFN-γ Levels in NC/Nga Mice

Increased levels of serum IgE are a well-known clinical feature of AD. Additionally, the levels of T helper 1 (Th1) and T helper 2 (Th2) cytokines associated with IgE production have been reported to correlate with the clinical severity of AD in NC/Nga mice [10,25]. The effects of DSMs on IgE levels and Th1/Th2 cytokine balance were investigated in AD mouse models. Serum IgE levels were elevated in the Dfb- or DNCB-induced AD groups (Figure 3A). Notably, serum IgE levels were significantly lower in AD mice treated with DSMs compared to those in the Dfb- or DNCB-treated AD mice (*p* < 0.05). The reduction in IgE levels observed with the cream formulation of DSMs appeared to be more pronounced than that of the mist formulation. The increase in Th2 cytokines that promote IgE production is a critical mediator in the progression of early AD. Interleukin-4 (IL-4), a Th2 cytokine, was elevated in the Dfb- or DNCB-induced AD mouse models compared to the control group (Figure 3B). The DSM-treated AD mice exhibited lower IL-4 levels than the Dfb- or DNCB-induced AD mice; furthermore, the cream formulation of DSMs demonstrated a stronger inhibitory effect than the mist formulation. Interestingly, interferon-gamma (IFN-γ) levels were significantly elevated in the serum of Dfb- or DNCB-induced AD mice but were reduced in the DSM-treated group (Figure 3C). These results indicate that DSMs contribute to the inhibition of AD lesion progression in vivo by decreasing IgE levels and modulating serum IL-4 levels in Dfb- or DNCB-induced AD mice, irrespective of Th1 polarization mediated by IFN-γ.

### 3.4. Effect of DSM Administration on Histological Features and Infiltration of T Cells on Dfb- or DNBC-Induced AD Skin Lesions

Figure 4 shows the H&E staining of back skin sections obtained from the normal control, the AD-induced, and the DSM-treated AD mice, respectively. NC/Nga mice, matched for age and maintained under standard SPF conditions, served as control mice with healthy skin, exhibiting no pathological features in the collected tissues. Epidermal hyperplasia was observed in the tissues obtained from each Dfb- or DNCB-induced AD mouse. Treatment with DSMs significantly reduced epidermal hyperplasia in the AD-induced groups. Furthermore, the reduction in epidermal hyperplasia was more pronounced with the cream formulation of DSMs compared to the mist formulation (Figure 4). Next, the effect of DSMs on T cell infiltration in AD-like skin lesions was investigated. Immunohistochemical analysis using an anti-CD4 antibody was conducted on Dfb- or DNCB-induced AD skin lesions, both with and without DSM treatment. T cell infiltration was notably more prominent in tissues obtained from Dfb- or DNCB-induced AD mice, while the administration of DSMs inhibited T cell infiltration. Additionally, the inhibitory effect of DSMs on T cell infiltration was more significant for the cream formulation than for the mist formulation (Figure 5). These results indicate that the DSM treatment of Dfb- or DNCB-induced AD-like skin lesions in NC/Nga mice effectively inhibits epidermal hyperplasia and T cell infiltration into skin lesions.

## 4. Discussion

AD is an inflammatory skin disease that accompanies most allergy and asthma patients, and its incidence has been rapidly increasing in recent years [2,26,27]. The condition is associated with complex etiological factors, including abnormal immune responses and inflammation driven by skin barrier defects and external triggers such as environmental or food allergens [2,26]. Therefore, studies using animal models are essential for identifying these intricate causes and developing effective therapies. Mouse models are predominantly utilized due to their cost-effectiveness and the ease of genetic manipulation compared to dogs and guinea pigs. Numerous mouse models have been developed, with the spontaneous generation of NC/Nga mice first reported in 1997 [7]. However, spontaneous NC/Nga mouse models typically exhibit late-onset AD and low incidence rates of the condition (<50%) [28]. House dust mite allergens are significant contributors to human AD. Fukuyama et al. [29] reported that AD induced by house dust mite exposure in NC/Nga mice closely mimics traditional human AD. Furthermore, it has been documented that treatment with *Dermatophagoides* mite extracts in general mice under SPF conditions induces AD-like skin lesions [10,11]. *Dermatophagoides farinae* is the most prevalent mite species found on the skin of AD patients and is recognized as the most potent allergen associated with AD. Consequently, animal models treated with *Dermatophagoides farinae* are considered valuable for elucidating the pathogenesis of AD [9,12]. DNCB is a cytotoxic and electrophilic benzene derivative that, when administered to NC/Nga mice, induces skin changes typically associated with human AD caused by airborne allergens. DNCB-treated mice are regarded as a model for AD due to the skin irritant properties of DNCB [13]. In this study, AD induction models utilizing either Dfb or DNCB were employed to investigate the effects of DSMs and to compare the cream and mist formulations of DSMs.

Numerous studies have indicated that minerals present in aqueous solutions have a positive impact on various skin diseases, including AD. Zinc plays a crucial role in epidermal hyperplasia and wound healing [30]. Some magnesium salts and mixtures of magnesium and calcium salts promote the healing of the skin barrier from damage [31]. Minerals such as sulfur, manganese, magnesium, and bicarbonate found in mineral springs and spring water are beneficial for skin conditions like AD [3,32]. Furthermore, the water from the Dead Sea, renowned for its rich mineral content, offers protection against UV-induced stress on human skin, enhances skin barrier function, increases skin hydration, and alleviates skin diseases, including inflammation associated with atopic dry skin [33,34]. More recently, research has demonstrated that deep sea water, which has a mineral composition similar to that of substances derived from the Dead Sea, helps protect the skin barrier, reduces inflammation, and improves various skin conditions, including AD [23,24]. Deep sea water has also been reported to exhibit a range of physiological activities, such as anti-diabetic, anti-hypercholesterolemic, anti-obesity, and anti-cancer effects [19,20,21,22]. Lee et al. [35] investigated the therapeutic effects and mechanisms of deep sea water in treating atopic skin inflammation in NC/Nga mice and HaCaT cells. However, the optimal concentration and biological mechanisms underlying the anti-inflammatory effects of Dead Sea water and deep sea water remain to be fully elucidated.

AD is characterized by skin inflammation, erythema, dryness, and itching, and effective treatments for AD are limited [36]. Typically, emollients that create a barrier on the skin, retain moisture, and protect against irritation, along with topical medications that alleviate skin inflammation, are employed to manage AD symptoms [37]. Emollients are complex formulations containing a variety of chemicals with antimicrobial, anti-itch, and anti-inflammatory properties, specifically designed to enhance the softness and suppleness of the epidermis [38]. Topical formulations for emollients or moisturizers for AD are generally more easily applied. The selection of the formulation for delivering pharmacological activity to the skin and optimizing its function is crucial. In this study, various formulations of DSMs, including mist and cream, were chosen for topical application in the treatment of AD. The resulting symptoms of AD after treatment with the different DSM formulations were compared to identify the most effective formulation. IgE is a class of antibodies that plays a crucial role in allergic reactions and is associated with type 1 hypersensitivity. In response to external antigens such as pollen and house dust mites, B cells secrete IgE [16]. The secreted IgE interacts with inflammatory cells and mast cells, inducing degranulation, which leads to the release of inflammatory mediators such as histamine and leukotrienes [2,16]. It has been reported that IgE levels increase in individuals with AD and that the extent of this increase correlates with the severity of the condition [39]. In our study, serum total IgE levels were significantly elevated in models of AD induced by Dfb or DNCB, while treatment with DSMs reduced these levels. Furthermore, treatment with the cream formulation of DSMs resulted in significantly lower serum IgE levels compared to treatment with the mist formulation (Figure 3).

Another important factor in the skin lesions associated with AD is the presence of inflammatory cytokines [40]. Cytokines are primarily produced by white blood cells, particularly helper T cells (CD4^+^). Helper T cells are classified into Th1 lymphocytes, which promote cell-mediated immunity, and Th2 lymphocytes, which mediate humoral immunity based on the functions of the secreted cytokines [41]. Th2 lymphocytes produce cytokines such as IL-4 and IL-10, which regulate IgE synthesis and contribute to inflammation in AD [42]. Treatment with Dfb or DNCB increased IL-4 production in NC/Nga mice; however, IL-4 production decreased following DSM treatment. This effect was more pronounced with the cream formulation of DSMs compared to the mist formulation. Interestingly, IFN-γ production also increased with Dfb or DNCB treatment (Figure 3). It is well established that atopic dermatitis is associated with the excessive secretion of IL-4 and IL-10 by Th2 cells, leading to the unstable production of IL-12 and the reduced secretion of IFN-γ [43]. However, it has been reported that in chronic AD, the Th1 immune response becomes more prominent [44]. Therefore, our results suggest that IFN-γ overproduction may increase as the disease progresses into a chronic state in the later stages of AD.

Our data demonstrate that treatment with DSMs in Dfb- or DNCB-induced AD-like lesions significantly reduces Th2 cytokines, serum IgE levels, and clinical severity scores. Moreover, the cream formulation of DSMs is more effective than the mist formulation. These results suggest that the topical application of DSMs, particularly in cream form, can effectively alleviate the symptoms of atopic dermatitis. Additionally, the topical application of DSMs may serve as an effective strategy for preventing other allergic skin diseases. Since this study has limited results in an animal model, additional research is needed on long-term administration studies or co-administration with other moisturizers that have skin barrier protection effects.

## 5. Conclusions

This study investigated the effects of DSMs in mist and cream formulations on AD-like skin lesions in NC/Nga mice. DSMs, especially in cream form, showed promising therapeutic effects by reducing AD-like lesion formation, trans-epidermal water loss, erythema, serum IgE, IFN-γ, and IL-4 levels, and skin thickness. They also reduced CD4^+^ T cell infiltration, suggesting a suppression of the Th2 immune response. The cream formulation outperformed the mist formulation. These results suggest that DSMs, especially in cream form, may be a suitable treatment for the relief and prevention of allergic dermatitis.

## Figures and Tables

**Figure 1 biomedicines-13-00861-f001:**
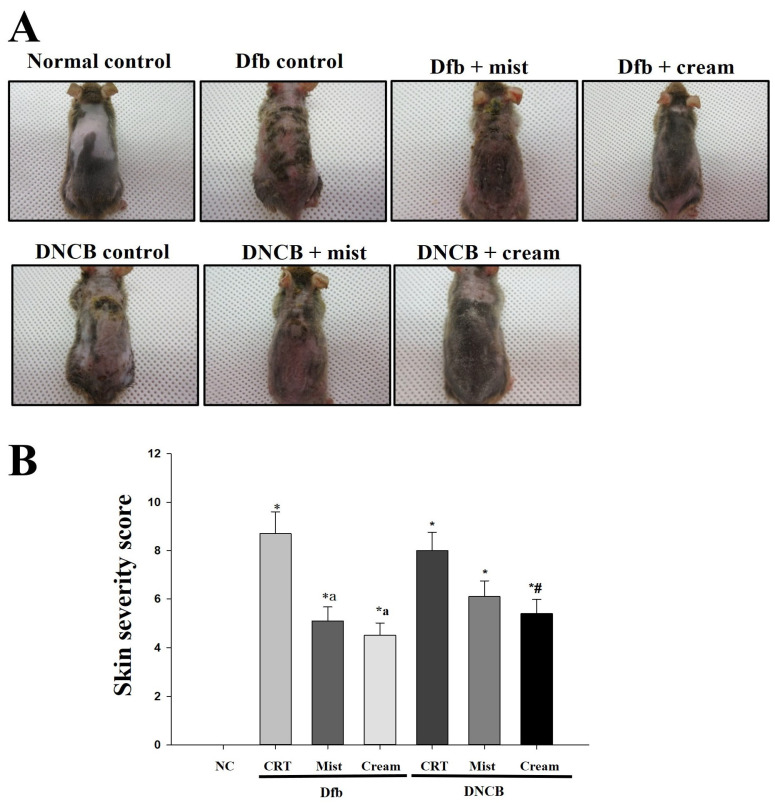
The effect of DSMs on Dfb- or DNCB-induced AD-like skin lesions in NC/Nga mice. (**A**) Images of the dorsal skin lesions on NC/Nga mice. (**B**) Skin severity scores. The scoring represents a clinical index. The scoring was evaluated on scarification day. Bars represent mean ± SEM. *: *p* < 0.05 vs. NC group; a: *p* < 0.05 vs. Dfb CRT group; #: *p* < 0.05 vs. DNCB CRT group. NC, normal control; CRT, control; Mist, DSM mist; Cream, DSM cream.

**Figure 2 biomedicines-13-00861-f002:**
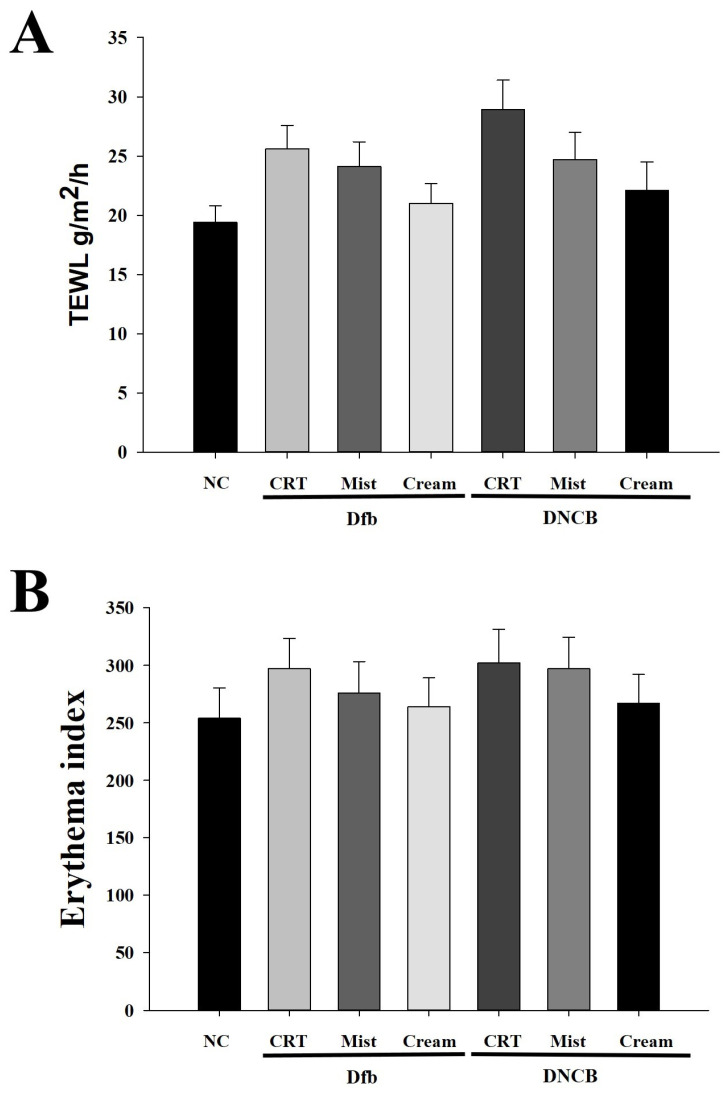
Effect of DSMs on biophysical parameters of the skin in Dfb- or DNCB-treated NC/Nga mice. (**A**) Trans-epidermal water loss (TEWL) in the mouse dorsal skin was measured using a skin evaporative water recorder in the sacrificed mice. (**B**) The epidermal thickness of the dorsal skin was measured from the stratum basale to the stratum granulosum (excluding the stratum corneum) using an image analysis system. Data represent the mean ± SEM (n = 5). NC, normal control; CRT, control; Mist, DSM mist; Cream, DSM cream.

**Figure 3 biomedicines-13-00861-f003:**
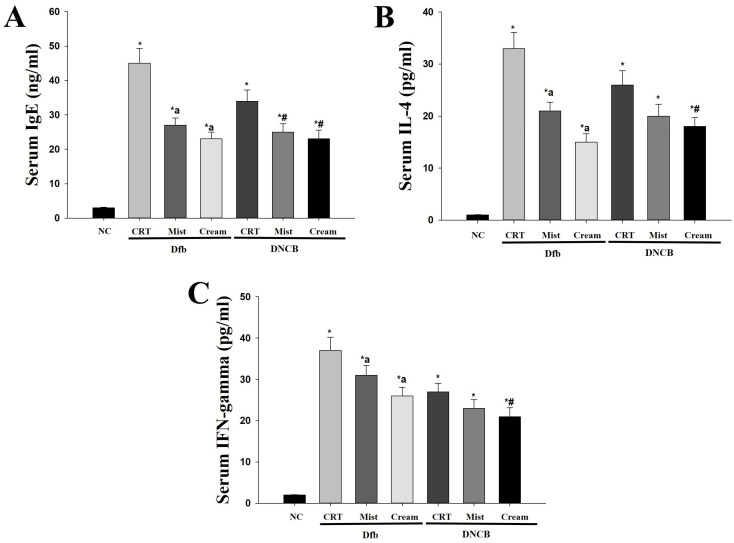
Determination of changes in IgE, IL-4, and IFN-γ serum levels due to treatment with DSMs in Dfb- or DNCB-treated NC/Nga mice. Levels of serum IgE (n = 5) (**A**), IL-4 (n = 5) (**B**), and IFN-γ (n = 5) (**C**) were measured by ELISA. Data represent mean values ± SEM (n = 5). Bars represent mean ± SEM. *: *p* < 0.05 vs. NC group; a: *p* < 0.05 vs. Dfb CRT group; #: *p* < 0.05 vs. DNCB CRT group. NC, normal control; CRT, control; Mist, DSM mist; Cream, DSM cream.

**Figure 4 biomedicines-13-00861-f004:**
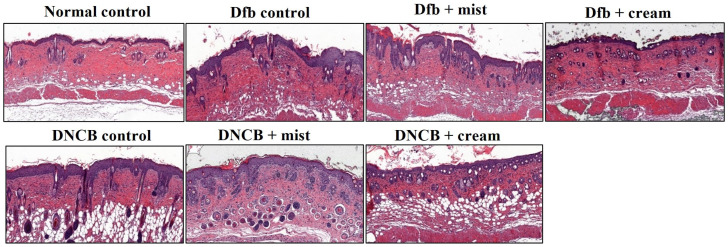
Effect of DSMs on the histological structure of Dfb- or DNCB-induced AD-like skin lesions in NC/Nga mice. The dorsal skin of untreated mice, Dfb-treated mice, DNCB-treated mice, Dfb plus mist formulation DSM-treated mice, Dfb plus cream formulation DSM-treated mice, DNCB plus mist formulation DSM-treated mice, and DNCB plus cream formulation DSM-treated mice was collected on day 14. Skin sections were stained with H&E for histological evaluation. Original magnification, ×400. Mist, DSM mist; Cream, DSM cream.

**Figure 5 biomedicines-13-00861-f005:**
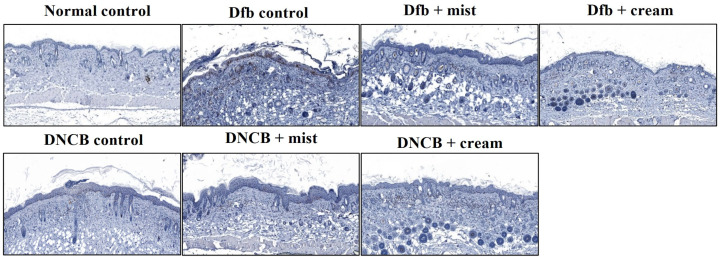
Effect of DSMs on Dfb- or DNBC-induced T cell infiltration into skin lesions. The dorsal skin of untreated mice, Dfb-treated mice, DNCB-treated mice, Dfb plus mist formulation DSM-treated mice, Dfb plus cream formulation DSM-treated mice, DNCB plus mist formulation DSM-treated mice, and DNCB plus cream formulation DSM-treated mice was collected on day 14. Skin lesions were stained with anti-CD4 antibody to identify T cells. T cells were displayed under a microscope at 400× magnification. Mist, DSM mist; Cream, DSM cream.

## Data Availability

Data are contained within the article.

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
