# Peer review of "Deep Sea Minerals Ameliorate Dermatophagoides Farinae- or 2,4-Dinitrochlorobenzene-Induced Atopic Dermatitis-like Skin Lesions in NC/Nga Mice"

_biomedicines, 2025, doi:10.3390/biomedicines13040861_

Round 1

Reviewer 1 Report

Comments and Suggestions for Authors

Unnecessary abbreviations in the text such as “..atopic dermatitis (AD),” “Atopic dermatitis (AD), characterized..”

“Seawater” is sometimes union sometimes not ..

The sentences of “Atopic dermatitis (AD), characterized by typical skin pruritus and erythema, is an inflammatory condition that affects at least 15% of children and is on the rise [2, 26]. Most patients with AD have a personal or family history of allergies or asthma [27].” Should be eliminated from Discussion section.

A proper should be added after “Magnesium ions (Mg²⁺) have been shown to inhibit antigen presentation by human epidermal Langerhans cells both in vivo and in vitro.”

Dead Sea ???

deep sea water (DSW),?

CD4+ in place of CD4+

Not suitable format an abbreviation and informing the reader in Conclusions section like “This study investigated the effects of deep-sea minerals (DSM) in mist and cream formulations on atopic dermatitis (AD)-like skin lesions in NC/Nga mice.”

If the proper explanation was not included in the Introduction section about the abbreviation, it should added, if so, abbreviations should be removed.

Author Response

Comments 1: Unnecessary abbreviations in the text such as “atopic dermatitis (AD),” “Atopic dermatitis (AD), characterized.”

Response 1: Thank you for pointing this out. We have revised it.

Comments 2: “Seawater” is sometimes union sometimes not.

Response 2: We have revised it. All seawater has been converted to sea water.

Comments 3: The sentences of “Atopic dermatitis (AD), characterized by typical skin pruritus and erythema, is an inflammatory condition that affects at least 15% of children and is on the rise [2, 26]. Most patients with AD have a personal or family history of allergies or asthma [27].” Should be eliminated from Discussion section.

Response 3: We have revised it. The alternative sentence was written as follows: AD is an inflammatory skin disease in which most patients are accompanied by allergies or asthma, and its incidence has increased rapidly in recent years [2,26,27].

Comments 4: A proper should be added after “Magnesium ions (Mg²⁺) have been shown to inhibit antigen presentation by human epidermal Langerhans cells both in vivo and in vitro.”

Response 4: We have revised it.

Comments 5: Dead Sea ???

Response 5: We have revised it. Dead Sea, which first appears in line 310, refers to the Dead Sea in the Middle East. As an example of seawater rich in minerals, there are research reports on its anti-inflammatory and skin-protecting effects.

Comments 6: deep sea water (DSW),?

Response 6: We have revised it. DSW removed.

Comments 7: CD4+ in place of CD4+

Response 7: We have revised it.

Comments 8: Not suitable format an abbreviation and informing the reader in Conclusions section like “This study investigated the effects of deep-sea minerals (DSM) in mist and cream formulations on atopic dermatitis (AD)-like skin lesions in NC/Nga mice.”

If the proper explanation was not included in the Introduction section about the abbreviation, it should added, if so, abbreviations should be removed.

Response 8: We agree with this comment. It has been corrected.

The abbreviations are explained in the introduction.

The rest are unified as abbreviations.

Changed from human atopic dermatitis (AD) to human AD in line 52.

Changed from deep-sea minerals  deep sea minerals (DSMs) in line 80.

Changed to ‘This study investigated the effects of DSMs in mist and cream formulations on AD-like skin lesions in NC/Nga mice.’ in line 370.

Reviewer 2 Report

Comments and Suggestions for Authors

The manuscript entitled "Deep sea minerals ameliorate Dermatophagoides farinae or 2,4-dinitrochlorobenzene-induced atopic dermatitis in NC/Nga mice"  The study addresses an important area of research regarding atopic dermatitis and potential therapeutic interventions. The use of NC/Nga mice as a model for atopic dermatitis is appropriate and well-justified. However there are some queries and comments that should be addressed by the authors before making it acceptable.

  1. Author should infer the ethical committe aprooval number in the materials and methods section.
  2. The experimental design appears robust, with appropriate controls in place. However, it would be beneficial to include a more detailed description of the dosing regimen for the deep sea minerals.
  3. The sample size used in the experiments should be clearly stated, and a power analysis should be included to justify the number of animals used.

  4. While the study demonstrates the efficacy of deep sea minerals in ameliorating symptoms, further exploration of the underlying mechanisms would strengthen the conclusions. Consider discussing potential pathways involved in the observed effects.
  5. The discussion could benefit from a more thorough comparison with existing literature on treatments for atopic dermatitis. How do the effects of deep sea minerals compare with other known treatments?
  6. The authors should address any limitations of the study, including potential confounding factors and the generalizability of the findings to human populations.
  7. Suggestions for future research could be included, such as exploring the long-term effects of deep sea minerals or their efficacy in other models of atopic dermatitis.
  8. The conclusion effectively summarizes the findings, but it could be strengthened by emphasizing the potential clinical implications of the research.
  9.  

Author Response

The manuscript entitled "Deep sea minerals ameliorate Dermatophagoides farinae or 2,4-dinitrochlorobenzene-induced atopic dermatitis in NC/Nga mice" The study addresses an important area of research regarding atopic dermatitis and potential therapeutic interventions. The use of NC/Nga mice as a model for atopic dermatitis is appropriate and well-justified. However there are some queries and comments that should be addressed by the authors before making it acceptable.

Comments 1: Author should infer the ethical committe aprooval number in the materials and methods section.

Response 1: Added IACUC approval number.

Comments 2: The experimental design appears robust, with appropriate controls in place. However, it would be beneficial to include a more detailed description of the dosing regimen for the deep sea minerals.

Response 2: Thank you for pointing this out. We have revised it. There are no standards for the use of deep sea minerals, but products that meet the quasi-drug approval standards of the Republic of Korea were used in this study. Deep sea mineral cream and deep sea mineral mist contain 5% deep sea minerals. The content of deep sea minerals is briefly described in 'Materials and Methods'.

Comments 3: The sample size used in the experiments should be clearly stated, and a power analysis should be included to justify the number of animals used.

Response 3: Thank you for pointing this out. We have revised it. Five mice were randomly assigned to each experimental group. Except for the normal control group, only animals that had atopic dermatitis were used in the experiment.

Comments 4: While the study demonstrates the efficacy of deep sea minerals in ameliorating symptoms, further exploration of the underlying mechanisms would strengthen the conclusions. Consider discussing potential pathways involved in the observed effects.

Response 4: Thank you for your comment. As a result of this paper, only the mechanisms related to IL-4 and IFN-r among cytokines are discussed. This study is part of the initial research to confirm the effect of deep sea minerals on improving dermatitis, and has limitations in that it did not confirm changes in IL-17, Th2 cytokines, etc. We also recognize that additional research is necessary, and we ask for your understanding on this point. Some potential pathways are mentioned in the ‘introduction’ and ‘discussion’.

Comments 5: The discussion could benefit from a more thorough comparison with existing literature on treatments for atopic dermatitis. How do the effects of deep sea minerals compare with other known treatments?

Response 5: We agree with this comment. Although studies have been reported on the improvement of skin diseases using Dead Sea water in dermatitis models and the skin barrier recovery effect of mineral complex materials, there are few cases of evaluating the efficacy of deep sea minerals as a single ingredient on the skin. Minerals can be administered in various ways, such as orally, transdermally, and by injection, so they can be applied more widely than existing treatments. This is also the reason for conducting this study.

Comments 6: The authors should address any limitations of the study, including potential confounding factors and the generalizability of the findings to human populations.

Response 6: Thank you for your comment. This study is an early stage study that confirmed the alleviating effect of minerals on mouse atopic dermatitis, and the results are insufficient to discuss the application to atopic dermatitis caused by complex and diverse causes in humans. This is also a limitation of animal experiments, and this limitation is noted in the paper.

Comments 7: Suggestions for future research could be included, such as exploring the long-term effects of deep sea minerals or their efficacy in other models of atopic dermatitis.

Response 7: We completely agree with your comment. Future research is expected to focus on long-term administration or concurrent administration with other skin-protecting moisturizers, and these have been added to the 'Discussion'.

Comments 8: The conclusion effectively summarizes the findings, but it could be strengthened by emphasizing the potential clinical implications of the research.

Response 8: We have revised it.

Reviewer 3 Report

Comments and Suggestions for Authors

Major Issues:

  • The abstract and introduction refer to "AD-like skin lesions" and "AD," yet the study involves the topical application of irritants or allergens (Dfb and DNCB). Is the paper actually about contact dermatitis—either irritant contact dermatitis (ICD) or allergic contact dermatitis (ACD)—or are the authors specifically discussing AD-like skin lesions? AD has a strong genetic component, and while irritants can exacerbate it, they do not directly cause AD. If ICD or ACD is the primary focus, all references to "AD" should be changed accordingly, including in the title.
  • Similar to the above point, the abstract incorrectly states that one can "induce AD."
  • The statement, "The underlying causes of AD include abnormal responses to medications, microbial infections, and environmental allergens," is inaccurate. Medications and allergens exacerbate AD due to an impaired barrier function; they do not cause it. The references cited (Elias et al. 2008 and Kaufman et al. 2018) primarily discuss immune system abnormalities and skin barrier dysfunction as central to AD pathogenesis.
  • The introduction mentions "skin lesions resembling those observed in human atopic dermatitis," but this requires further clarification. AD-like lesions and true AD are different than each other.
  • The phrase "allergic antigens" in the introduction is appropriate for ACD and AD-like lesions, but not ICD, which results from direct contact with irritants rather than allergens.
  • The discussion states, "Dermatophagoides mite extracts in general mice under SPF conditions induce AD," which is incorrect. The references cited (Matsuoka et al. 2003 and Sasakawa et al. 2001) discuss atopic eczema/dermatitis syndrome (AEDS) and AD-like lesions, respectively. Furthermore, Masahiro et al. 1999 describes NC/Nga mice as a model for AD, noting they develop "AD-like skin lesions," not AD itself.
  • Since this is an animal study, a dedicated paragraph on the pathophysiology of ICD, ACD, or AD-like skin lesions in mice is necessary.
  • The paper states that the mice "were assessed by shaving the hair from the back of the mice." What method was used for shaving? The technique (epilation vs. depilation) can influence skin integrity and potentially confound lesion formation. Was hair removed by waxing or mechanical epilation (which removes the bulb), or by shaving or depilatory cream (which leaves the bulb intact)? The authors should clarify this.
  • In Figure 1, the "normal control" mouse appears to be photographed immediately post-shaving, while other mice exhibit some hair regrowth. What phase of hair growth were they in? Were all images taken on the same post-shave day? Standardizing image capture timing is necessary for accurate comparisons.
  • A new figure illustrating the experimental timeline would improve clarity. The current methods section is confusing—on what days were Dfb and DNCB applied? When were DSM mist and cream administered? When was the skin harvested?
  • In Figure 1, "DNCB" alone should be labeled as "DNCB control."
  • In Figure 2, specify on which day the mouse skin was harvested.
  • In Figure 1, explain how TEWL and erythema thickness were measured, similar to Figure 2.
  • The methods section should be moved directly after the introduction.
  • The described experimental groups in the methods do not match the provided images in Figure 1a. Ensure the figures match these designations.
    • The text states: “Group 1: untreated normal control (NC), Group 2: Dfb ointment-induced AD as a positive control (DfbPC), Group 3: Dfb ointment + DSMC (DfbC), Group 4: Dfb ointment + DSMM (DfbM), Group 5: DNCB-induced AD as a positive control (DNCBPC), Group 6: DNCB + DSMC (DNCBC), Group 7: DNCB + DSMM (DNCBM)"
  • The authors should discuss why this specific mouse model is suitable for dermatitis research. What other studies have used this model for dermatitis, eczema, or related skin diseases?
  • If the study focuses on ICD or ACD instead of AD-like lesions, an alternative clinical scoring system for skin severity may be needed.

Minor Issues:

  • Once abbreviation is introduced, do not revert to the full term.
  • All figure legends should define abbreviations (e.g., "CTR" means "control"). Instead of just "mist" and "cream," use "DSM mist" and "DSM cream" for clarity.
  • In the results section (2.4), define "SPF" in line 159.
  • Dfb should be identified as house dust mite antigen in lines 295-296
Comments on the Quality of English Language

The paper can be edited by a native English speaker to improve its flow and clarity.

Author Response

Comments 1: The abstract and introduction refer to "AD-like skin lesions" and "AD," yet the study involves the topical application of irritants or allergens (Dfb and DNCB). Is the paper actually about contact dermatitis—either irritant contact dermatitis (ICD) or allergic contact dermatitis (ACD)—or are the authors specifically discussing AD-like skin lesions? AD has a strong genetic component, and while irritants can exacerbate it, they do not directly cause AD. If ICD or ACD is the primary focus, all references to "AD" should be changed accordingly, including in the title.

Response 1: Thank you for your comment. The following is information about the animal models we investigated in preparation for this study: The mouse atopic dermatitis model is used to study the pathogenesis and treatment of atopic dermatitis using laboratory mice that exhibit symptoms similar to human atopic dermatitis. There are various types of mouse models, each with different characteristics and advantages and disadvantages.

The NC/Nga mice used in this study naturally exhibit symptoms similar to atopic dermatitis. It has the advantage of being able to observe the natural pathogenesis similar to human atopic dermatitis, but it is known that the timing of onset and the severity of symptoms vary from individual to individual.

DNCB is known to be a substance that can induce atopic dermatitis by repeatedly applying it to mouse skin, but the stimulation by the inducing substance may differ from the actual pathogenesis of atopic dermatitis.

As you pointed out, the mouse model does not perfectly reflect human atopic dermatitis, but it is known to be a model that can sufficiently screen for effects on basic dermatitis. In most animal studies, species differences are considered when interpreting experimental results.

Comments 2: Similar to the above point, the abstract incorrectly states that one can "induce AD."

Response 2: Thank you for pointing this out. Same as above answer.

Comments 3: The statement, "The underlying causes of AD include abnormal responses to medications, microbial infections, and environmental allergens," is inaccurate. Medications and allergens exacerbate AD due to an impaired barrier function; they do not cause it. The references cited (Elias et al. 2008 and Kaufman et al. 2018) primarily discuss immune system abnormalities and skin barrier dysfunction as central to AD pathogenesis.

Response 3: Thank you for your valuable comments. We know that medications can cause immune system abnormalities. Antibiotics such as penicillin and cephalosporins, NSAIDs such as ibuprofen and aspirin, anticonvulsants such as phenytoin and carbamazepine, biological agents such as TNF inhibitors, antifungal agents, antiviral agents, and some blood pressure medications are known to cause allergic reactions, causing rashes and itching similar to atopic dermatitis. Our experiment was conducted to compare the effects of DSMs according to its formulation in NC/Nga mice, a well-known animal model similar to human AD, and although it has some similarities to human atopic dermatitis, there are limitations in using animal models for a complete comparison.

Comments 4: The introduction mentions "skin lesions resembling those observed in human atopic dermatitis," but this requires further clarification. AD-like lesions and true AD are different than each other.

Response 4: We agree with your opinion. In the introduction, it was stated that this experiment has limitations in that it is an animal experiment and cannot completely show human AD.

Comments 5: The phrase "allergic antigens" in the introduction is appropriate for ACD and AD-like lesions, but not ICD, which results from direct contact with irritants rather than allergens. The discussion states, "Dermatophagoides mite extracts in general mice under SPF conditions induce AD," which is incorrect. The references cited (Matsuoka et al. 2003 and Sasakawa et al. 2001) discuss atopic eczema/dermatitis syndrome (AEDS) and AD-like lesions, respectively. Furthermore, Masahiro et al. 1999 describes NC/Nga mice as a model for AD, noting they develop "AD-like skin lesions," not AD itself.

Response 5: Thank you for your comment. Modified to AD-like skin lesions.

Comments 6: Since this is an animal study, a dedicated paragraph on the pathophysiology of ICD, ACD, or AD-like skin lesions in mice is necessary.

Response 6: Thank you for your comment. We performed clinical scoring of AD-like skin lesions based on references 7, 15, and 16.

Comments 7: The paper states that the mice "were assessed by shaving the hair from the back of the mice." What method was used for shaving? The technique (epilation vs. depilation) can influence skin integrity and potentially confound lesion formation. Was hair removed by waxing or mechanical epilation (which removes the bulb), or by shaving or depilatory cream (which leaves the bulb intact)? The authors should clarify this.

Response 7: Thank you for pointing this out. Hair removal was performed with animal clippers, and considering skin irritation, DNCB and Dfb were applied after a 1-day recovery period after shaving. Materials and methods were supplemented.

Comments 8: In Figure 1, the "normal control" mouse appears to be photographed immediately post-shaving, while other mice exhibit some hair regrowth. What phase of hair growth were they in? Were all images taken on the same post-shave day? Standardizing image capture timing is necessary for accurate comparisons.

Response 8: Thank you for your comment. All experimental groups, including the normal control, were photographed 4 weeks after shaving. The normal control group's photo is presented to show the shaving conditions. In experiments, the hair growth status after shaving varies greatly among animals. It has been changed to a different photo.

Comments 9: A new figure illustrating the experimental timeline would improve clarity. The current methods section is confusing—on what days were Dfb and DNCB applied? When were DSM mist and cream administered? When was the skin harvested?

Response 9: Thank you for suggesting the experimental timeline. This study is relatively simple and is identical to the methods in other references, so it has been modified in more detail. Please understand.

Comments 10: In Figure 1, "DNCB" alone should be labeled as "DNCB control."

Response 10: Figure 1 has been revised.

Comments 11: In Figure 2, specify on which day the mouse skin was harvested.

Response 11: We have revised it. Transepidermal water loss (TEWL) of the dorsal skin was measured 4 weeks after application of DSMs using VapoMeter (Delfin Technologies Ltd., Kuopio, Finland).

Comments 12: In Figure 1, explain how TEWL and erythema thickness were measured, similar to Figure 2.

Response 12: Thank you for your comment. Figure 1 has been revised. Figure 1 shows the macroscopic evaluation after 4 weeks of application of DSM mist and cream formulations. The photographs and results in Figures 1 and 2 are the results measured on the day the animals were sacrificed, 4 weeks after the application of DSMs. These contents have been revised.

Comments 13: The methods section should be moved directly after the introduction.

Response 13: Thank you for your guidance. The Materials and Methods section has been moved.

Comments 14: The described experimental groups in the methods do not match the provided images in Figure 1a. Ensure the figures match these designations.

The text states: “Group 1: untreated normal control (NC), Group 2: Dfb ointment-induced AD as a positive control (DfbPC), Group 3: Dfb ointment + DSMC (DfbC), Group 4: Dfb ointment + DSMM (DfbM), Group 5: DNCB-induced AD as a positive control (DNCBPC), Group 6: DNCB + DSMC (DNCBC), Group 7: DNCB + DSMM (DNCBM)"

Response 14: Thank you for your valuable comments. We have revised the Materials and Methods.

Comments 15: The authors should discuss why this specific mouse model is suitable for dermatitis research. What other studies have used this model for dermatitis, eczema, or related skin diseases?

Response 15: Thank you for your guidance. The following is information about the animal models we investigated while preparing this study.

DNCB-induced model: DNCB is a chemical widely used to induce atopic dermatitis, and this model induces skin lesions similar to human atopic dermatitis. DNCB is directly applied to the skin to stimulate the immune response, inducing the activation of Th2 cells and the secretion of inflammatory cytokines. It is used in various studies and is useful for evaluating the efficacy of treatments. Many studies have contributed to understanding the pathological mechanism of atopic dermatitis and developing new treatments through the DNCB model (Characterization of Different Inflammatory Skin Conditions in a Mouse Model of DNCB-Induced Atopic Dermatitis. Inflammation 47, 771–788 (2024). Establishment and Characterization of Mild Atopic Dermatitis in the DNCB-Induced Mouse Model, Int J Mol Sci. 2023 Aug 1;24(15)).

House dust mite model: House dust mites, which are allergens, are one of the main causes of atopic dermatitis, and this model simulates the mechanism of atopic dermatitis development in a real environment. Atopic dermatitis can be induced using mites such as Dermatophagoides farinae (A mouse model of the atopic eczema/dermatitis syndrome by repeated application of a crude extract of house-dust mite Dermatophagoides farina. Allergy. 2003 Feb;58(2):139-45.) Therefore, the DNCB and house dust mite models are widely used as models suitable for atopic dermatitis research.

Comments 16: If the study focuses on ICD or ACD instead of AD-like lesions, an alternative clinical scoring system for skin severity may be needed.

Response 16: Thank you for your thoughtful comments. We will continue to consider this part in the future. ICD is considered to be induced cell death, and ACD is considered to be activated induced cell death. In general, most AD-like skin lesions are characterized by itching, erythema, excoriation, scaling, and dryness. Since this study focuses on AD-like skin lesions, no other clinical scores are required.

Comments 17: Once abbreviation is introduced, do not revert to the full term.

Response 17: The entire abbreviation has been revised.

Comments 18: All figure legends should define abbreviations (e.g., "CTR" means "control"). Instead of just "mist" and "cream," use "DSM mist" and "DSM cream" for clarity.

Response 18: We have revised Figure legends.

Comments 19: In the results section (2.4), define "SPF" in line 159.

Response 19: In the introduction, it is already defined as specific pathogen-free conditions (SPF).

Comments 20: Dfb should be identified as house dust mite antigen in lines 295-296

Response 20: Thank you for pointing this out. The sentence was broken as "Two substances were used to experimentally induce AD. DNCB and house dust mite antigen." It has been corrected to "Two substances, DNCB and house dust mites (Dfb), have been used to experimentally induce AD."

Round 2

Reviewer 1 Report

Comments and Suggestions for Authors

accept as is

Comments on the Quality of English Language

accept as is

Author Response

We sincerely appreciate your valuable review.

Reviewer 2 Report

Comments and Suggestions for Authors

The manuscript is revised well by the authors as per the review comments pointed out by me during first revision. I agree with the revision stage and I give my recommendation for publication in the present form.

Author Response

(The authors gave the same response as above.)

Reviewer 3 Report

Comments and Suggestions for Authors

I would like to thank the authors for their hard work.

However, they need to incorporate their responses to the manuscript, in particular: #1, #3, #6, #15 and #16.

Author Response

Comments 1: The abstract and introduction refer to "AD-like skin lesions" and "AD," yet the study involves the topical application of irritants or allergens (Dfb and DNCB). Is the paper actually about contact dermatitis—either irritant contact dermatitis (ICD) or allergic contact dermatitis (ACD)—or are the authors specifically discussing AD-like skin lesions? AD has a strong genetic component, and while irritants can exacerbate it, they do not directly cause AD. If ICD or ACD is the primary focus, all references to "AD" should be changed accordingly, including in the title.

Response 1: Thank you for your comment. As is well known, the etiology of human AD involves heterogeneous and complex factors such as genetic, immune, environmental, and microbial imbalances, and there is currently no mouse model that perfectly mimics human AD. However, for various studies including the development of therapeutics for AD, a mouse animal model is inevitably required. The NC/Nga mouse spontaneous AD model used in this study is widely known to be similar to the clinical characteristics of human AD. It is reported to be particularly useful for studies on immune responses. However, it has limitations because the modeling period is long and unstable. Therefore, in order to evaluate efficacy in preclinical trials, excellent stability and reproducibility in the onset and severity of skin lesions must be demonstrated, so an induced model using irritants such as DCNB and Dfb was combined. Of course, these irritant models have limitations similar to the contact dermatitis model, but as mentioned in the main text of the paper, models using DCNB and Dfb in NC/Nga mice have been traditionally used as animal models for evaluating the efficacy of therapeutics for treating human AD. For this reason, AD was used in the title. In this study, if we had applied irritants or allergens such as Dfb and DNCB to inbred mice such as Balb/C or B6, which are not natural models, we would have used the title by distinguishing between ICD and ACD. However, as you know, NC/Nga mice are a spontaneous AD model that shows AD-like skin lesions. In consideration of your review, we have changed the title from 'atopic dermatitis' to 'atopic dermatitis-like skin lesions'.

The following are references on NC/Nga mice as an AD model.

1. Ewald D.A., Noda S., Oliva M., Litman T., Nakajima S., Li X., Xu H., Workman C.T., Scheipers P., Svitacheva N., Labuda T., Krueger J.G., Suárez-Fariñas M., Kabashima K., Guttman-Yassky E. Major differences between human atopic dermatitis and murine models, as determined by using global transcriptomic profiling. J. Allergy Clin. Immunol. 2017;139:562–571. 

2. Jin H., He R., Oyoshi M., Geha R.S. Animal models of atopic dermatitis. J. Invest. Dermatol. 2009;129:31–40.

3. Gittler J.K., Shemer A., Suárez-Fariñas M., Fuentes-Duculan J., Gulewicz K.J., Wang C.Q.F., Mitsui H., Cardinale I., de Guzman Strong C., Krueger J.G., Guttman-Yassky E. Progressive activation of T(H)2/T(H)22 cytokines and selective epidermal proteins characterizes acute and chronic atopic dermatitis. J. Allergy Clin. Immunol. 2012;130:1344–1354.

4. Matsuda H., Watanabe N., Geba G.P., Sperl J., Tsudzuki M., Hiroi J., Matsumoto M., Ushio H., Saito S., Askenase P.W., Ra C. Development of atopic dermatitis-like skin lesion with IgE hyperproduction in NC/Nga mice. Int. Immunol. 1997;9:461–466.

5. Ye S, Zhu L, Ruan T, Jia J, Mo X, Yan F, Liu J, Zhang Y, Chen D. Comparative study of mouse models of atopic dermatitis. Heliyon. 2025 Jan 18;11(2):e41989.

Comments 3: The statement, "The underlying causes of AD include abnormal responses to medications, microbial infections, and environmental allergens," is inaccurate. Medications and allergens exacerbate AD due to an impaired barrier function; they do not cause it. The references cited (Elias et al. 2008 and Kaufman et al. 2018) primarily discuss immune system abnormalities and skin barrier dysfunction as central to AD pathogenesis.

Response 3: Thank you for your valuable comments. It has been corrected to " The underlying causes of AD include genetics, immune dysregulation, epidermal dysfunction, and environmental factors.”.

Comments 6: Since this is an animal study, a dedicated paragraph on the pathophysiology of ICD, ACD, or AD-like skin lesions in mice is necessary.

Response 6: Thank you for your comment. We performed clinical scoring of AD-like skin lesions based on references 7, 15, and 16. Additionally, the pathophysiology of AD-like skin lesions was mentioned in the introduction section while describing NC/Nga mice.

Comments 15: The authors should discuss why this specific mouse model is suitable for dermatitis research. What other studies have used this model for dermatitis, eczema, or related skin diseases?

Response 15: Thank you for your guidance. The following is information about the animal models we investigated while preparing this study.

DNCB-induced model: DNCB is a chemical widely used to induce atopic dermatitis, and this model induces skin lesions similar to human atopic dermatitis. DNCB is directly applied to the skin to stimulate the immune response, inducing the activation of Th2 cells and the secretion of inflammatory cytokines. It is used in various studies and is useful for evaluating the efficacy of treatments. Many studies have contributed to understanding the pathological mechanism of atopic dermatitis and developing new treatments through the DNCB model (Characterization of Different Inflammatory Skin Conditions in a Mouse Model of DNCB-Induced Atopic Dermatitis. Inflammation 47, 771–788 (2024). Establishment and Characterization of Mild Atopic Dermatitis in the DNCB-Induced Mouse Model, Int J Mol Sci. 2023 Aug 1;24(15)).

House dust mite model: House dust mites, which are allergens, are one of the main causes of atopic dermatitis, and this model simulates the mechanism of atopic dermatitis development in a real environment. Atopic dermatitis can be induced using mites such as Dermatophagoides farinae (A mouse model of the atopic eczema/dermatitis syndrome by repeated application of a crude extract of house-dust mite Dermatophagoides farina. Allergy. 2003 Feb;58(2):139-45.) Therefore, the DNCB and house dust mite models are widely used as models suitable for atopic dermatitis research.

The above content is partially included in the 'Discussion'.

Comments 16: If the study focuses on ICD or ACD instead of AD-like lesions, an alternative clinical scoring system for skin severity may be needed.

Response 16: Thank you for your thoughtful comments. We will continue to consider this part in the future. In general, most AD-like skin lesions are characterized by itching, erythema, excoriation, scaling, and dryness. Since this study focused on AD-like skin lesions, we do not think that a comparison of clinical scores with other similar diseases is necessary.

Round 3

Reviewer 3 Report

Comments and Suggestions for Authors

The authors have addressed all comments.